# The Extracellular Metabolome Stratifies Low and High Risk Potentially Premalignant Oral Keratinocytes and Identifies Citrate as a Potential Non-Invasive Marker of Tumour Progression

**DOI:** 10.3390/cancers13164212

**Published:** 2021-08-21

**Authors:** Lee Peng Karen-Ng, Emma Louise James, Abish Stephen, Mark Henry Bennett, Maria Elzbieta Mycielska, Eric Kenneth Parkinson

**Affiliations:** 1Centre for Oral Immunobiology and Regenerative Medicine, Institute of Dentistry, Barts and the London School of Medicine and Dentistry, Queen Mary University of London, Turner Street, London E1 2AD, UK; karennlp@um.edu.my (L.P.K.-N.); dremmajames@gmail.com (E.L.J.); a.s.stephen@qmul.ac.uk (A.S.); 2Oral Cancer Research & Coordinating Centre (OCRCC), Faculty of Dentistry, University of Malaya, Kuala Lumpur 50603, Malaysia; 3Department of Life Science, South Kensington Campus, Imperial College London, London SW7 2AZ, UK; mark.h.bennett@btinternet.com; 4Department of Surgery, University Medical Center, Franz-Josef-Strauß Allee 11, 93053 Regensburg, Germany; Maria.Mycielska@klinik.uni-regensburg.de

**Keywords:** oral premalignancy, tumour progression, metabolism, senescence, diagnostics, tumour heterogeneity

## Abstract

**Simple Summary:**

The early detection of oral cancer is a high priority, as improvements in this area could lead to greater cure rates and reduced disability due to extensive surgery. Oral cancer is very difficult to detect in over 70% of cases as it develops unseen until quite advanced, sometimes rapidly. Therefore, the development of markers in body fluids (liquid biopsies) indicative of cancerous changes have a high priority. We show here that small molecules called metabolites can distinguish between non-diseased oral cells and two types of cells found in oral cells on the road to cancer. Although our investigation is preliminary, some of the metabolites have already been detected in the saliva (split) of oral cancer patients, and could eventually help detect oral cancer development at an earlier stage.

**Abstract:**

Premalignant oral lesions (PPOLs) which bypass senescence (IPPOL) have a much greater probability of progressing to malignancy, but pre-cancerous fields also contain mortal PPOL keratinocytes (MPPOL) that possess tumour-promoting properties. To identify metabolites that could potentially separate IPPOL, MPPOL and normal oral keratinocytes non-invasively in vivo, we conducted an unbiased screen of their conditioned medium. MPPOL keratinocytes showed elevated levels of branch-chain amino acid, lipid, prostaglandin, and glutathione metabolites, some of which could potentially be converted into volatile compounds by oral bacteria and detected in breath analysis. Extracellular metabolites were generally depleted in IPPOL, and only six were elevated, but some metabolites distinguishing IPPOL from MPPOL have been associated with progression to oral squamous cell carcinoma (OSCC) in vivo. One of the metabolites elevated in IPPOL relative to the other groups, citrate, was confirmed by targeted metabolomics and, interestingly, has been implicated in cancer growth and metastasis. Although our investigation is preliminary, some of the metabolites described here are detectable in the saliva of oral cancer patients, albeit at a more advanced stage, and could eventually help detect oral cancer development earlier.

## 1. Introduction

Oral cancer, predominantly squamous cell carcinoma (OSCC), is the sixth most common cancer worldwide and the management of this cancer has barely improved in decades. One of the problems in treating OSCC is that they frequently develop from a field of genetically and phenotypically diverse cells that are histologically undetectable [1].

Senescence can be induced by a variety of cellular stresses, but is bypassed in normal keratinocytes by the dual reduction in p53 and p16^INK4A^ function and telomerase deregulation. Genetic alterations in all of the above pathways are very common in OSCCs in vivo and in immortal cell lines derived from potentially pre-malignant lesions (PPOL) and OSCC [2,3,4,5]. 

Advanced cancers, and even some high-risk PPOL (HR-PPOL), have extensive gene copy number variations [6] and develop genetic heterogeneity and, consequently, the platform for drug resistance [7]. Recent evidence suggests that this genetic heterogeneity [8] and the induction of cellular senescence [9] develops prior to most OSCCs becoming visible [8] and immortal keratinocytes with extensive gene copy number variations have been demonstrated in PPOLs of low histological grade [5]. Furthermore, it has long been known that even advanced OSCCs contain both mortal and immortal keratinocytes [10,11], sometimes mixtures of the two [11], and the same is true of PPOLs [5,12]. 

Despite the fact that both mortal (MPPOL) and immortal (IPPOL) keratinocytes have neoplastic-like phenotypes [10], and altered transcriptional [12] and metabolic profiles, MPPOL have no gene copy number variations or gene methylation, few classical ‘driver’ mutations [13], and similar keratinocytes have been detected in OSCC [5,10,12,13]. 

MPPOLs and IPPOLs may often be completely distinct lesions, as their transcriptional profiles appear to diverge upon progression to mortal and immortal OSCC keratinocytes, although in some instances MPPOLs may be precursor lesions of IPPOLs [14,15,16]. As IPPOL and HR PPOLs have extensive gene copy number variations [6], loss of heterozygosity [17], and p16^INK4A^ dysfunction [5], the breakdown of cellular senescence in PPOL is likely important in progression to OSCC; this is supported by data from mouse models of other cancers [18]. There is also evidence that PPOL with features of IPPOL have a much higher risk of progression to OSCC than MPPOLs (HR IPPOL and LR MPPOL, respectively [17,19,20]).

However, senescent cells secrete an array of proteins that have important functions in age-related diseases and cancers, known as the senescence-associated secretory phenotype (SASP) [21]; recent data suggests that senescent keratinocytes secrete some SASP factors [22] and metabolites in vitro and, in mouse models of epidermal SCC, in vivo [23,24]. Therefore, LR MPPOL keratinocytes may well have a tumour-promoting role in the pre-cancerous field of OSCC.

In addition, LR MPPOL and mortal OSCC are associated with a different class of cancer-associated fibroblasts [25]. Whilst a considerable amount is known about cancer-associated fibroblasts and their role in modulating carcinoma behaviour, including OSCC [26], far less is known about the different types of keratinocytes that exist within cancerous and pre-cancerous fields and how they may influence one another’s behaviour.

Several groups have attempted to discriminate between PPOL and OSCC by the metabolomic analysis of saliva and serum, with the objective of developing non-invasive strategies for the early detection of OSCC. Whilst this objective is laudable, the previous studies have so far been frustrated by variable collection criteria and a variety of platforms used to analyse the metabolites [27,28,29]. Signatures of PPOL and OSCC have been reported recently [27], but it is not yet clear how many of these metabolites are related to the PPOL keratinocytes or inflammatory disease [28]. Additionally some previously published PPOL metabolites may be due to bacterial breakdown products of other metabolites, and changes in the oral microbiome are known to occur in PPOL and OSCC [30]. To date, in no case have the PPOL lesions been characterised for markers that distinguish HR IPPOL from LR MPPOL. Therefore, the relationship of the published work to the different types of PPOL keratinocytes, and thus treatment or preventative strategies, is not presently clear. 

We have used a panel of well characterised PPOL keratinocyte cultures which represent the different stages of PPOLs on the road to immortality and aneuploidy in vivo; these cultures have been defined genetically [2,5,10,13], and transcriptionally [12]. We have recently characterised the extracellular metabolites of senescent fibroblasts [31] and have used the same strategy to identify the extracellular metabolites of M- and HR IPPOL keratinocytes cultures with the aim of distinguishing them from each other and from normal oral keratinocytes. 

Here, we report extracellular metabolites that discriminate LR MPPOL and HR IPPOL from each other and their normal counterparts, such as breakdown products of the branch chain amino acid pathway, lipids, a ketone, E prostaglandins, and gamma-glutamyl amino acids and some candidate amino acids for conversion to volatile compounds by oral bacteria. Significantly, several of the metabolic changes that distinguish HR IPPOL from LR MPPOL are reported to distinguish OSCC from PPOL in human saliva [27]. Far fewer metabolites were upregulated in the HR IPPOL group, but one of them, citrate, has important functions in cancer growth and metastasis [32,33]. The results are discussed in relation to their potential to minimally invasively detect and classify oral dysplasia and the potential of others to promote OSCC progression.

## 2. Materials and Methods

### 2.1. Cell Culture

The normal and PPOL human keratinocytes were grown using lethally irradiated 3T3 feeders as described previously [5,10] in Dulbecco’s Modified Eagles Medium (DMEM) supplemented with 10% vol/vol foetal bovine serum (FBS—Hyclone foetal clone II), 0.4 μg/mL hydrocortisone, 2 mM glutamine (Gibco Life technologies, ThermoFisher Scientific, Dartford, Kent, UK) and antibiotics (streptomycin 50 U/mL Penicillin and 50 µg/mL Streptomycin) lacking citrate buffer (Sigma, Poole, Dorset, UK). The cultures were maintained in an atmosphere of 10% CO_2_/90% air and sub cultured once weekly at a density of 1 × 10^5^ cells per 9 cm plate to prohibit confluence. 

### 2.2. Cell Lines Used in the Study

The characteristics and properties of the LR- and IPPOL line used in the study are given in Appendix A.

### 2.3. Ethical Approval

Ethical approval for the PPOL lines (with informed consent) was granted by the Glasgow Dental Hospital Area Ethics Committee (10MAR97/AGN4vi) and the Edinburgh Dental Hospital Area Ethics Committee (before 1995) and for the normal NHOK keratinocytes by Central and South Bristol Research Ethics Committee Project E5133: Cell proliferation, differentiation, and apoptosis in oral squamous cell carcinoma.

### 2.4. Collection of the Conditioned Medium for Analysis of the Extracellular Metabolites

The collection of the conditioned medium and cell pellets was carried out essentially as described previously [31,34], except that the keratinocytes were allowed to reach confluence before the collection began, to more accurately represent the state of the epithelium in vivo. As lethally irradiated 3T3 cells would undoubtedly contribute to the extracellular metabolome of the NHOK and PPOL cultures, the cultures were allowed to grow to around 50% confluence in dishes before removing the 3T3 feeders with 0.02% EDTA and vigorous pipetting and washing 3 times with calcium- and magnesium-free phosphate buffered saline (PBS). The keratinocytes were then removed and plated in such a way as to reach confluence within 72 h in T25 flasks, before adding 3 mL fresh medium to each flask for 24 h and snap freezing the fresh medium at zero time, to elucidate the background and also whether metabolites accumulated or were depleted [31,34]. The cells were counted by rinsing with 0.02% EDTA and disaggregating the cells with 0.1% trypsin and 0.01% EDTA. The trypsin was neutralized with an excess of DMEM and FBS and the cells counted on a haemocytometer. The cell densities plated to achieve confluence in 72 h were as follows: D19, 1.44 × 10^5^; D35, 1.6 × 10^5^; DOK, 7.2 × 10^4^; NHOK810, 4 × 10^4^; D6, 5.5 × 10^4^; D9, 1.06 × 10^5^; E4, 3.2 × 10^4^; D25, 8 × 10^4^; D4, 5.5 × 10^3^; D20, 8.3 × 10^5^; D34, 6.67 × 10^4^; and D17, 1.33 × 10^5^. All cultures were originally tested for mycoplasma at the time of isolation [5] and found to be negative, and this has been confirmed recently for D6, D25, D4, D19, D20, D34, D35, OKF4, and OKF6 by others [7,35], and their identity confirmed by short tandem repeat profiling [35] or their molecular characteristics [14].

### 2.5. Senescence-Associated Beta Galactosidase (SA-β Gal) Assay

The 3T3 feeders were removed from sub-confluent keratinocyte cultures as stated above, rinsed with PBS, fixed for 10 min at room temperature and stained for 24 h for SA-β Gal using a commercial kit from (Biovision K320, Biovision, Milpitas, CA, USA) according to the manufacturer’s instructions.

### 2.6. Antibodies and Western Blotting

The antibodies used were p16-^INK4A^ (10883-1-AP) rabbit polyclonal, 1 in 5000 (Proteintech, Manchester, UK); rabbit monoclonals from Abcam, Cambridge, UK anti-MCM7 [EP1974Y] (ab52489) at 1 in 10,000 and anti-SIRT1 [E104] (ab32441) at 1 in 20,000; rabbit polyclonal Anti-β-actin (ab8227) at 1 in 20,000 (Abcam, Cambridge, UK); rabbit monoclonal GAPDH (14C10) (#2118) at 1 in 4000 (Cell Signaling Technology, Beverly, MA, USA) The dilution used for each antibody was optimised to obtain clear band on the positive controls.

Cell pellets were thawed on ice before adding 50–100 µL radio immunoprecipitation assay (RIPA) buffer (Sigma, Dorset, UK) with protease and phosphatase inhibitors and lysed for 30 min at 4 °C. Following removal of cell debris by centrifugation at 12,000 rpm for 20 min at 4 °C, the protein, quantitated by the Bradford Assay and total cellular protein, was separated based on molecular weight on 4–12% gradient sodium dodecyl sulphate polyacrylamide gels under denaturing and reducing conditions. Following protein transfer, the nitrocellulose membrane was blocked with 5% wt/vol milk protein prepared in Tris Buffer Saline and Tween 20 (TBS-T) for 1 h at room temperature (RT). The primary antibodies were diluted in 5% wt/vol milk protein in TBS-T and the membrane/blot was incubated/probed overnight with primary antibody at 4 °C, washed 3 times in TBS-T for 5 min at room temperature under agitation. The membranes were incubated with appropriate immunoglobulin G horseradish peroxidase (IgG HRP)—conjugated secondary antibody diluted as above for 1 h at RT. Antigen-antibody complexes were detected by incubating with ECL Western Blotting Substrate for 1 min or for sensitive detection, ECL Prime Western Blotting Detection Reagent or SuperSignal^®^ West Fem to Maximum Sensitivity Substrate according to manufacturer’s protocols. Membranes were exposed to the Amersham Hyperfilm ECL and developed in the dark using a standard film developer machine. Densitometry analysis was performed on scanned films using Image J. The relative intensities of the bands of interest were normalised against the values obtained from the corresponding loading controls.

### 2.7. Metabolomic Analysis, Normalisation, and Data Presentation as Scaled Intensity

The metabolomic screen was carried out by Metabolon Inc. Morrisville, NC 27560, who also prepared Figures 1–5. The details of the metabolomics analysis have been published previously, including sample preparation, instrumentation, conditions for mass spectrometry (liquid chromatography/tandem mass spectrometry in positive and negative ion modes, and gas chromatography/mass spectrometry), peak data reduction, and assignment of peaks to known chemical entities by comparison to metabolite library entries of purified standards [31,34]. Briefly, for analysis, the median of a given biochemical was determined across all sample groups. This median was subsequently used to scale individual samples to a median of 1 for the group. A minimum value was assigned when, rarely, a biochemical was not detected in an individual sample. This approach takes into account day-to-day variations in instrument performance, but preserves differences between the experimental groups. The data is graphically presented as scaled intensity, and is thus a measure of the relative level of each metabolite of the experimental groups within each experiment. The scaled intensities were either not normalised, normalized to account for the variation in biomass or cell number between different experimental groups, or were normalized for cell number following subtraction of the medium blank values, and expressed as net scaled intensity per 10^5^ cells per mL of conditioned medium. Normalising for cell number did not change the pattern of the results dramatically as the cultures were confluent but, as D35 cells were somewhat smaller than the others, normalising for cell number did reduce the values in this cell line by around 50%. The reason for presenting the results in two ways was that non-normalised data was more comparable to the clinical setting of equal surface area, but normalizing to cell number gave additional insight into potential mechanism.

### 2.8. Targeted Measurement of Extracellular Citrate by Gas Chromatography/Mass Spectroscopy (GCMS)

Deuterated citric acid (2,2,4,4, d4 citric acid) from Sigma Aldrich, Poole Dorset, UK was added to each sample to a final concentration of 0.1 mM as an internal standard. Metabolites were then extracted using cold methanol before being dried under vacuum desiccation. The samples were re-suspended in anhydrous pyridine containing the derivatisation agents methoxyamine hydrochloride, followed by N-Methyl-N-trimethylsilyltrifluoroacetamide, with 1% 2,2,2-Trifluoro-N-methyl-N-(trimethylsilyl)-acetamide, and Chlorotrimethylsilane (MSTFA + 1%TMCS). GCMS was performed in pulsed splitless mode on a Hewlett Packard HP6890 series GC system with Agilent 6890 series injector, a 30 m long 250 µm diameter capillary column (Agilent, Stockport, Cheshire, UK) model number 19091s-433HP5MS) using a flow rate of 1 mL/min, and a Hewlett Packard 5973 Mass selective detector. The acquisition was conducted in selective ion monitoring mode, the ion masses detected for citrate were: 273, 347, 375, and 465 and the corresponding heavy ions were 276, 350, 378, and 469. The dwell time for all these ions was 50 ms. Data were normalised for cell number following subtraction of the medium blank value, and expressed as mM citrate per 10^5^ cells per mL.

### 2.9. Statistical Analysis 

A Welch’s *t*-test two-sample was used to identify biochemicals that differed significantly between experimental groups in the unbiased metabolomic screen. Pathways were assigned for each metabolite, allowing examination of overrepresented pathways. Additionally, citrate in the experimental groups was analysed where indicated, by the Wilcoxon–Mann–Whitney Test. All data were additionally analysed by one-way analysis of variance to test the difference between multiple samples, and where appropriate by Student’s unpaired *t*-test. All data were based on a minimum of 3 independent experiments per cell line unless otherwise stated.

## 3. Results

### 3.1. Characteristics of the Cell Line Panel

The properties of the PPOL lines used in the study and the clinical details of the patients from whom they were derived are given in Appendix A. The PPOL lines (Appendix A) have been transcriptionally profiled [12] and extensively characterised phenotypically and genetically [2,5,10,13]. The LR MPPOL lines D6/D30 are diploid, mortal in vitro, regulate telomerase normally, and have intact senescence effectors p16^INK4A^ and p53, whereas the HR IPPOL lines are aneuploid, immortal in vitro, have deregulated telomerase, and generally lack the senescence effectors p16^INK4A^ and p53 [5]. D17 may be on the road to immortality in that it has an extended replicative lifespan and lacks p16^INK4A^ expression [5] but has functional p53 [5,16] and regulates telomerase normally. HR IPPOL lines D19, D20, and D35 were of additional interest as the patients from which they were derived from progressed to OSCC within 5 years of being biopsied [12]. The patients from which DOK [36], D4, and D17 were derived had metachronous OSCC lesions (Appendix A). Two normal oral keratinocytes cultures, NHOK810 and NHOK881, were also analysed. We chose to investigate the PPOL panel in the 3T3 support system originally validated to culture several SCC samples optimally [37] as this approach leads to the multiplication and maintenance [11] of keratinocytes harbouring the same genetics and EGFR expression patterns of both PPOL and SCC-HN in vivo [5,13,38,39].

### 3.2. Overview of the Unbiased Metabolomic Screen

Conditioned media from several confluent normal and PPOL keratinocytes cultures was analysed as previously described for fibroblasts [31]. The raw data from the screen is shown in Appendix A. The data presented in Figure 1, Figure 2, Figure 3, Figure 4, Figure 5 and Figure 6 were from cultures of equal surface area and not normalised for cell number or protein content, to permit comparisons with published human body fluid signatures, where such normalization was not possible. However, full metabolite lists of both non-normalised and those still significant following subtraction of the background in fresh medium and normalization for cell number are shown in Appendix A. Principal component analysis (Figure 1A) revealed a distinct separation between sample types, suggesting differences in the extracellular metabolic profiles of these keratinocyte cell lines. Notably, the two LR MPPOL media samples (D6, D30) and the rapidly progressing HR IPPOL media sample (D35) exhibited significant separation from the other lines. Hierarchical clustering (Figure 1B) also consistently sorted the majority of samples by class. The LR MPPOL samples (D6 and D30) samples tightly clustered together and may be indicative of a shared biochemical signature.

### 3.3. Metabolites Potentially Relevant to Keratinocyte Senescence

Interestingly, separation was observed between the normal keratinocyte groups NHOK810 and NHOK881, and may reflect differences between these cell types. Indeed, NHOK881 was more senescent than NHOK810 as assessed by senescence-associated beta galactosidase (SA-β Gal) staining, and failed to reach complete confluence. Therefore, we also compared NHOK810 and NHOK881 to gain a preliminary insight into the potential role of normal keratinocyte senescence on the data.

Appendix A shows the metabolites which differed between NHOK810 and NHOK881 without normalisation, and the same data with the fresh medium control were subtracted and the conditioned medium values corrected for cell number (Appendix A). Prior to normalization for background and cell number, significantly higher levels of cystine, 4-hydroxyphenylpyruvate and certain lipids were significantly elevated in the more senescent NHOK881, and many metabolites depleted, especially those connected with leucine, isoleucine and valine metabolism, gamma-glutamyl amino acids, and the ketone 3-hydroxybutyrate (Appendix A). However, following correction for background and cell number (Appendix A) only the mono-hydroxy fatty acid 3-hydroxybutyrate and pyrimidine metabolites thymine and uracil were statistically elevated. The lysolipid 2-oleoylglycerophosphocholine* and 3-aminobutyrate were significantly depleted. Nevertheless, the results suggest that the inclusion of NHOK881 in the non-normalised data may need to be taken into account; therefore, only metabolites that retained statistical significance following normalization are highlighted below.

Similar to the previously published microarray analysis [12], the LR MPPOL and HR IPPOL keratinocytes could easily be separated from NHOK on the basis of their extracellular metabolomes, and were even more different from each other. Interestingly, many of the metabolites that accumulated in the conditioned medium of the LR MPPOL keratinocytes relative to that of NHOKs were the same as those found in the fibroblast extracellular senescence metabolome, raising the question of whether the LR MPPOLs were more senescent than the NHOKs or the HR IPPOLs. However, Western blot analysis of an extended panel of LR MPPOL keratinocytes showed that they had no more p16^INK4A^ expression than normal oral keratinocyte line NHOK810 (Figure 2A), and all the HR IPPOLs lacked p16^INK4A^ expression, as published previously [5,39]. The LR MPPOL keratinocytes continued to express sirtuin 1 (SIRT1), a highly specific marker of senescence in oral fibroblasts [34] (Figure 2B). However, LR MPPOL keratinocytes do show increased SA-β Gal staining (Figure 2D), lower levels of MCM2/7 (Figure 2C), and expression of some but not all SASP cytokine transcripts [12], suggesting that LR MPPOL keratinocytes are showing the early stages of senescence. Therefore, the altered extracellular metabolites in LR MPPOL relative to normal could be due to the early stages of keratinocyte senescence and their ablation in HR IPPOL following immortalization. In many instances, the altered HR PPOL metabolites were observed following the partial breakdown of senescence in line D17 which still senesces in vitro despite lacking p16^INK4A^ expression [5].

#### 3.3.1. Branched Chain Amino Acids

Compared to fresh media, the branched chain amino acids (BCAA) valine, isoleucine, and leucine were reduced in conditioned cell media samples and may be indicative of uptake and utilization for protein synthesis and/or energy metabolism. Indeed, LR MPPOL D6/D30 samples possessed slightly elevated levels of the alpha-keto acids 4-methyl-2-oxopentanoate, 3-methyloxobutyrate, and 3-methyl-2-oxovalerate, and strikingly increased levels of the related downstream degradation products such as isovalerate and 3-hydroxyisobutyrate compared to NHOK controls (Appendix A; Figure 3)**.** These trends were not consistently observed in D17 and the HR IPPOL (D4/D9/D19/D20/D35) culture media (Appendix A; Figure 3), and may reflect a defining difference in BCAA catabolism that is potentially influenced by altered metabolism following the breakdown of senescence.

#### 3.3.2. Lipid Metabolism 

Culture media from LR MPPOL D6/D30 keratinocytes possessed higher levels of multiple long chain fatty acids including palmitate, palmitoleate, margarate, 10-heptadecenoate, and oleate (Appendix A; Figure 4) compared to NHOK controls. Higher levels of ethanolamine and choline were also observed in LR MPPOL D6/D30 media, coupled with lower phospholipid degradation products (Appendix A). Additionally, D6/D30 media possessed elevated levels of the ketone body 3-hydroxybutyrate (BHBA). In contrast, the HR IPPOL keratinocytes exhibited significantly lower levels of BHBA compared to NHOK controls (Appendix A; Figure 4). Both long chain fatty acids and polyunsaturated fatty acid levels were significantly reduced in the five HR IPPOL keratinocyte media compared to NHOK control and D6/D30 samples (Appendix A; Figure 4).

#### 3.3.3. Prostaglandin Metabolism

Prostaglandins are oxidized essential fatty acids that are generated by the cyclooxygenase pathway and contribute to the regulation of physiological processes such as inflammation, differentiation, and vasoconstriction. Elevated levels of multiple polyunsaturated fatty acids including linoleate, linolenate, and docosapentaenoate in LR MPPOL (D6/D30) samples (Appendix A; Figure 5) suggested increased substrate availability for eicosanoid synthesis. In support, D6 and particularly D30 media exhibited higher levels of prostaglandin (PG) E2, A2, and E1 compared to NHOK control samples (Appendix A; Figure 5). Aside from eicosanoids, elevated levels of the lipid peroxidation products 13-HODE and 9-HODE were observed in LR MPPOL and D20 media (Figure 5). In the HR IPPOL media, PGEs and PGA2 were generally lower or undetectable (Appendix A; Figure 5).

#### 3.3.4. Glutathione Metabolism 

Differences in lipid peroxidation levels between media samples suggested that redox homeostasis may also be altered between the different keratinocytes groups. Compared to NHOK controls, four out the five HR IPPOL lines analysed (D4, D9, D20, and D35) media possessed elevated levels of oxidized (GSSG) glutathione (Appendix A; Figure 6) that may reflect increased free radical exposure. Notably, reduced glutathione (GSH) levels were also elevated in these samples (Appendix A; Figure 6) and may suggest increased biogenesis from the rate limiting metabolite cysteine as potentially suggested by lower levels in D4 and D35 media (Figure 6), although this was not observed in the media of D9 and D20. Although GSH and GSSG levels were below the limit of detection in D6/D30 media (Appendix A; Figure 6), multiple gamma-glutamyl amino acids including gamma-glutamylmethionine and gamma-glutamylphenylalanine were elevated in these samples relative to normal (Appendix A; Figure 6). A similar trend was not observed in all five HR IPPOL samples and the gamma-glutamyl amino acid catabolite 5-oxoproline was not significantly altered between sample groups.

#### 3.3.5. Other Metabolites

Several other metabolites are significantly elevated in LR MPPOL keratinocytes when compared to normal, including several involved in sterol, amino acid, purine and pyrimidine metabolism (Appendix A). However, in the HR IPPOL keratinocyte media, only four metabolites other than oxidized and reduced glutathione (described above) showed a statistically significant increase over normal keratinocytes. Dimethylarginine (SDMA + ADMA) showed a 2.93-fold increase (Appendix A), but showed a very similar increase in LR MPPOL keratinocyte conditioned media (3.1-fold, Appendix A). Most metabolites were depleted in HR IPPOL keratinocytes relative to normal, including glutamate, branch chain amino acid metabolites, long chain and polyunsaturated fatty acids, phospholipid metabolism purine metabolism, and pyrimidine metabolism (Appendix A). However, three metabolites, homocysteine (6.45-fold), citrate (3.66-fold), and N1-methyladenosine (2.11-fold) were specifically and significantly elevated relative to normal in the conditioned media of the HR IPPOL group but not the LR MPPOL group (Appendix A). Citrate, (4.88-fold), N1-methyladenosine (2.27-fold), oxidized glutathione (26.77-fold), and gulono-1,4-lactone (2.94-fold) were also significantly elevated relative to normal in the media of the rapidly progressing HR IPPOL keratinocytes (D19, D20, and D35; Appendix A). Thus, these last four extracellular metabolites have potential as non-invasive diagnostic markers of HR IPPOLs if the basis of their specific elevation in HR IPPOL keratinocytes can be elucidated in the future.

### 3.4. Metabolites Distinguishing HR IPPOL and LR MPPOL and Their Relationship to Progression to OSCC In Vivo

All of the elevated metabolites also distinguished between HR IPPOL and LR MPPOL keratinocytes (Appendix A) but most distinguishing metabolites between these groups were depleted in HR IPPOL keratinocyte media compared to their LR MPPOL counterparts. The depleted metabolites included many metabolites of the branch chain amino acid pathway and long chain fatty acids as well as glycine and lysine metabolism (Appendix A). Most of these metabolites also distinguished HR IPPOL keratinocytes from both LR MPPOL and normal keratinocytes (Appendix A).

### 3.5. Relationship of the HR IPPOL and LR MPPOL Metabolomes to PPOL and SCC Saliva Signatures In Vivo

Interestingly, 19/33 metabolites recently reported to distinguish PPOL and OSCC in saliva in vivo [27] also showed similar trends between HR IPPOL and LR MPPOL keratinocytes. These included elevated 4-hydroxybutyrate, serine, glutamate, glycerol, and leucine, and depleted hippurate, proline, glycerol-3-phosphate, caprylate, histidine, glycerophosphoryl choline, arginine, tryptophan, creatine, and phenylalanine, which distinguished HR IPPOL from LR MPPOL in vitro; nine of these were significantly different (Figure 7). Depleted proline and caprylate remained significant when adjusted for background and cell number. Of the 14 metabolites distinguishing OSCC from PPOL saliva in vivo [27] which did not show the same trend as HR IPPOL versus LR MPPOL, 12 did show a similar trend when LR MPPOL were compared with NHOK. Six of the 12 were statistically significant, including palmitate, oleate, linoleate, indoleacetate, methionine, and uridine. Homocysteine, which has also been reported in PPOL patient saliva [29] was strikingly and significantly elevated in HR IPPOL conditioned medium compared to LR MPPOL and normal keratinocyte media (Figure 7 and Appendix A). 

### 3.6. Metabolites That Are Potentially Convertible into Volatile Compounds by Oral Bacteria

Excess extracellular amino acids produced by HR IPPOL and LR MPPOL cells may be metabolized by the resident oral microbiota, releasing volatile organic compounds (VOCs) as by-products. Amino acids such as cysteine, methionine, and tryptophan, and peptides such as glutathione are fermented by major commensal bacteria to produce volatiles such as hydrogen sulfide, methanethiol, and indole [40]. A number of proteolytic and inflammophilic genera have been shown to be associated with premalignant and malignant oral lesions, with some species being also hypothesized to be tumorigenic [41]. Exogenous fatty acids are also known to be utilized by the oral microbiota to be incorporated into membrane lipids, with elevated levels of fatty acids such as eicosenoate, oleate, and 3-hydroxypalmitate observed in comparisons between normal keratinocytes and LR MPPOL [42]. These provide clues to changes in the oral microenvironment from normal keratinocytes to LR MPPOL or HR IPPOL that may encourage enrichment of inflammophilic and proteolytic bacteria. In addition, a number of metabolites were identified, including those that are products of the leucine, isoleucine, and valine metabolism with vapour pressures that suggest they may be volatile and potentially be detected in the oral cavity headspace (Figure 8). These metabolites, and their enthalpy of vaporisation, are listed in Appendix A.

### 3.7. Extracellular Citrate (EC) Is Consistently Elevated in the Conditioned Medium of HR IPPOL Keratinocytes Relative to Normal and LR MPPOL Keratinocytes

Citrate has additional significance, as it has been implicated in the growth [43] and metastasis [32] of several cancer types and its transporter pmCiC is a potential pharmacological target (see [44,45] for reviews). Therefore, we studied this metabolite in more detail. 

In the unbiased metabolomics screen, citrate was elevated in the conditioned medium of HR IPPOL keratinocytes D4, D19, and D35, as well as D17, but was undetectable in NHOK and the LR MPPOL lines D6 and D30 (Figure 9A) and the same pattern was observed when the data was normalized for cell number (Figure 9B). We then studied an extended panel of LR MPPOL and HR IPPOL keratinocytes, and analysed citrate by targeted analysis using gas chromatography and mass spectroscopy (Figure 9C). The results reinforced the conclusions of the unbiased screen and showed that all eight HR IPPOL lines had detectable citrate, whilst only D30 of the LR MPPOL had any, and all eight HR IPPOL had more than this (*p* = 0.006).

## 4. Discussion

We observed several striking alterations in the extracellular metabolites of LR MPPOL keratinocytes cultures when compared with normal and HR IPPOL oral keratinocytes and these are summarized in the graphic. 

There were slight depletions of the BCAAs and a correspondingly slightly elevated levels of the alpha-keto acids and strikingly increased levels of their related downstream degradation products in the LR MPPOL, which were much less apparent in the HR IPPOL group. The accumulation of these metabolites may suggest enhanced catabolism and may contribute succinyl-CoA and acetyl-CoA to replenish the TCA cycle. 

Long chain fatty acid levels accumulated in LR MPPOL keratinocyte media more than in NHOK media, which may be indicative of enhanced lipid synthesis to support growth. Although these observations may alternatively implicate complex lipid hydrolysis, no significant differences were observed in multiple monoacylglycerols, suggesting complex lipid hydrolysis may not be altered. Furthermore, higher levels of ethanolamine and choline coupled with lower levels of the phospholipid degradation products glycerophosphorylcholine and glycerol 3-phosphate were observed in the LR MPPOL group compared to NHOK controls, and may reflect a shift in phospholipid metabolism to support membrane biogenesis. Elevated relative levels of the ketone body 3-hydroxybutyrate (BHBA) in LR MPPOL may reflect excess acetyl CoA levels that often reflect enhanced lipid oxidation. In contrast, HR IPPOL keratinocytes exhibited significantly lower levels of BHBA compared to NHOK controls that may be indicative of diminished β-oxidation highlighting limited lipid availability and supporting this, both long chain fatty acids and polyunsaturated fatty acid levels were significantly reduced in the HR IPPOL keratinocyte media compared to NHOK control and LR MPPOL samples.

PGE1, PGE2, and sometimes PGEA2 were elevated in LR MPPOL conditioned medium relative to that of the NHOKs, and largely absent (D4 excepted) in the HR IPPOL group. Aside from eicosanoids, elevated levels of the lipid peroxidation products 13-HODE and 9-HODE in LR MPPOL (Figure 5) may reflect oxidative stress and serve as peroxisome proliferator-activated receptor ligands in the LR MPPOL keratinocytes, which have high levels of PGEs 1 and 2. High levels of oxidative stress are known to be associated with certain types of cellular senescence and particularly important in keratinocytes [46]. Interestingly, PGE2 and 13,14-dihydro-15-keto-prostaglandin A2 were elevated in the more senescent NHOK881 relative to NHOK810 but PGE1 was not, indicating the specific regulation of PGE2 in oral keratinocyte senescence. 

The LR MPPOL keratinocytes possessed elevated levels of several gamma-glutamyl amino acids, which were essentially normal in the media of the HR IPPOL group. However, instead, strikingly elevated levels of oxidized and reduced glutathione relative in HR IPPOL lines compared to the other two groups were observed. The elevated levels of reduced glutathione in the HR IPPOL keratinocytes may suggest increased biogenesis from the rate limiting metabolite cysteine, which was depleted in some of the HR IPPOL cultures. Although cysteine depletion was not ubiquitous, strikingly elevated levels of the cysteine precursor homocysteine were observed in the media of most of the HR IPPOL keratinocytes. This could indicate elevated S-adenosyl synthetase activity to produce S-adenosyl methionine (SAM) which is also related to redox homeostasis [34]. The enzyme gamma-glutamyl transferase (GGT) catalyses the transfer of a gamma-glutamyl moiety of glutathione to an acceptor (an amino acid) and releases cysteinylglycine to provide cysteine for de novo glutathione synthesis. Consequently, these metabolites serve to facilitate the exchange of intra- and extracellular glutathione. In contrast, the gamma-glutamyl amino acid catabolite 5-oxoproline was not significantly altered between sample groups, suggesting that import and degradation may be similar between cell types. Thus, these findings may be indicative of enhanced GGT activity and are in agreement with evidence in the literature demonstrating GGT activity has value as a marker for preneoplastic changes in the oral epithelium [47]. However, these previous studies did not discriminate between LR MPPOLs and HR IPPOLs, the latter of which are at a higher risk of progression to malignancy.

Many of the extracellular metabolic changes associated with LR MPPOLs were similar to those observed for fibroblasts induced to senesce by irreparable DNA double strand breaks [31]. Therefore, we tested whether the LR MPPOL group was more senescent than normal keratinocytes and found that this was not the case as assessed by p16^INK4A^ levels. However, other markers of senescence, such as reduced proliferation, increased SA-β Gal staining, and some SASP cytokines were evident, suggesting that LR MPPOLs were either undergoing cellular stress or were in the early stages of senescence.

Extracellular metabolites that accumulate in HR IPPOL relative to both LR MPPOL and normal oral keratinocytes have the greatest diagnostic potential in detecting the former in saliva samples. Extracellular citrate cleanly distinguished between HR IPPOL, LR MPPOL, and normal oral keratinocytes in conditioned media. 

Recently, there has been considerable progress in the early detection of cancers in liquid biopsies such as serum and saliva [48,49]. However, even curable early stage oral cancer is expensive to treat and is often associated with unacceptable morbidity. Detecting HR IPPOL non-invasively is challenging but some of the metabolites described have been reported in studies of PPOL and OSCC in vivo. Elevated homocysteine and threonine which was associated with HR IPPOL keratinocytes and the depletion of the BCAA isoleucine, which was associated with LR MPPOL, distinguished oral leukoplakia from normal subjects in a study of saliva metabolites [29]. Significantly, the BCAA pathway was significantly elevated in LR MPPOL keratinocytes and depressed in the HR IPPOL keratinocytes media relative to their normal counterparts. 

A very recent study using conductive polymer spray ionization mass spectrometry which can profile saliva metabolites in seconds reported several metabolites that can non-invasively separate PPOL from normal healthy donors and OSCC [27]. Whilst this is encouraging some of the metabolites reported by Song et al. [27] have also been reported to be elevated in periodontal disease [28]. Some aspects of saliva metabolite signatures may be due to inflammation or bacterial infiltration, which has been reported and characterised for both PPOL and OSCC [30]. Nevertheless, of the observed elevated and depleted extracellular metabolites reported to distinguish OSCC from PPOL detectable in our study 60% showed a similar trend in distinguishing HR IPPOL from LR MPPOL in vitro and 30% were significant. Although 40% the remainder showed the opposite trend they did distinguish LR PPOL from normal keratinocytes and this may reflect the complex nature of neoplastic oral tissue. Alternatively, some of the changes that mediate the transition from PPOL to OSCC may not be related to keratinocyte status. No saliva study has yet attempted to stratify their PPOL subjects into high and low risk or characterized them beyond histological analysis and the metabolites reported here have the potential to stratify PPOL into low and high-risk lesions.

Of added interest we identified several metabolites (ketone bodies and members of the BCAA pathway) in LR MPPOL conditioned medium that could be converted by oral bacteria into volatile compounds that would potentially be detectable in the breath of high risk patient groups such as smokers or former smokers, alongside metabolites produced by the cells that may also be volatile. It will be interesting to test whether this is the case by testing the interaction of these metabolites with different types of oral bacteria known to be associated with PPOL and OSCC.

Unfortunately, citrate has not been investigated in studies of OSCC and PPOL saliva to date [27,28,29], as it also has functional significance in cancer [33]. In addition to the production of citrate by HR IPPOL it is also produced by senescent oral fibroblasts [31] and cancer-associated fibroblasts [32] which are also associated with OSCC [25]. Citrate has been implicated in tumour growth [43] and progression [32]. The plasma membrane citrate transporter pmCiC which shows altered expression and/or location in several malignancies, may be responsible for citrate uptake and increased cancer growth [33] and metastasis [32]. Furthermore, pmCiC has potential as a novel anti-cancer drug target [43,44,45]. Interestingly, citrate was reported to be depleted in a metabolomic study of head and neck SCC sera [50] which supports a role for citrate in fueling OSCC development and/or progression.

In addition, PGEs 1 and 2, which are strongly elevated in LR MPPOL keratinocytes are known to have several important functions relevant to OSCC. PGE2 production in oral cancer cells is often indicative of increased COX-2 expression which has been reported in PPOL in vivo [51], while excess PGEs and the PGE2 metabolite, PGA2, have been reported to have positive or negative effects on immunosurveillance depending on the setting [52,53,54]. Elevated levels of PGE1 can support metastasis as a potent endogenous vasodilator agent that increases peripheral blood flow and angiogenesis [55]. Considering prostaglandins can also suppress the humoral and cellular immune action responsible for the killing of malignant cancer cells [56,57], these observations may suggest increased pathogenicity in LR MPPOLs. LR MPPOLs can secrete numerous SASP factors [12] in addition to PGEs and have a tumour promoting function in a pre-cancerous field [23,24]. In addition, PGE2 can also stimulate the immune system in certain situations [56].

We acknowledge that the data reported here was obtained from keratinocytes cultured in a manner that does not allow complete differentiation, and that traits may have altered in vitro. However, the cell line panel described here allows variables and regulators to be investigated in the future and has the strength of providing a profile of normal and PPOL keratinocytes free of bacteria and other cell types. The cell line panels also have potential utility of screening for drugs that target LR MPPOL and/or HR IPPOL keratinocytes, and the extracellular metabolites described here and elsewhere [27,28,29] may help monitor their effectiveness in vivo.

## 5. Conclusions

Here, we have identified several extracellular metabolites, some of which are volatile, that distinguish LR MPPOL keratinocytes from HR IPPOL and many of these distinguish OSCC from PPOL in vivo in agreement with the extensive data showing that cellular immortality is important for tumour progression. In addition, metabolites, such as prostaglandins from LR MPPOL and citrate from HR IPPOL, are known to have biological roles in tumour growth and progression, suggesting interaction between the two PPOL classes.

The keratinocytes used in this study, although carrying similar genetic alterations to PPOLs and OSCCs in vivo, may have undergone extra alterations in vitro, and were cultured in a system that does not allow complete keratinocyte differentiation.

## Figures and Tables

**Figure 1 cancers-13-04212-f001:**
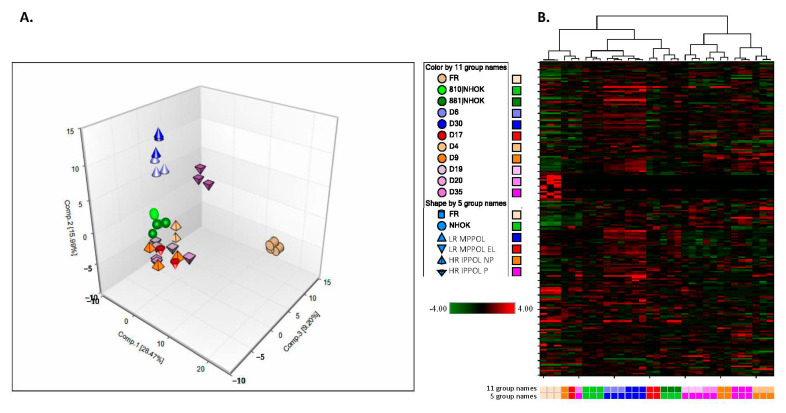
Principal component and cluster analysis of normal and PPOL keratinocyte extracellular metabolites. (**A**) Principal component analysis (PCA)—Low-risk mortal potentially premalignant oral lesion (LR MPPOL—D6 and D30), and rapidly progressing high-risk immortal potentially premalignant oral lesion (HR IPPOL P—D35) media samples exhibited significant separation from the other conditions analysed. The data from the individual cell lines and the fresh media (FR) are colour coded. The symbols represent into which group the individual lines are classified (normal human oral keratinocyte (NHOK—NHOK810 and NHOK881), LR MPPOL (D6 and D30), LR MPPOL extended lifespan (LR MPPOL EL—D17), HR IPPOL non-progressing (HR IPPOL NP—D4 and D9), or rapidly progressing (HR IPPOL P—D19, D20, and D35). The data show a clear separation of the LR MPPOL lines D6 and D30 (blue upright cones, circular base) and the HR IPPOL-P line D35 (dark purple inverted cones, square base) from the other groups and from each other. (**B**) Hierarchical clustering also consistently sorted the majority of the samples by cell class. The colour codes are the same as in (A). Interestingly, separation was observed between the normal keratinocyte groups NHOK810 (light green) and NHOK881 (dark green), the latter of which failed to reach confluence and appeared more senescent.

**Figure 2 cancers-13-04212-f002:**
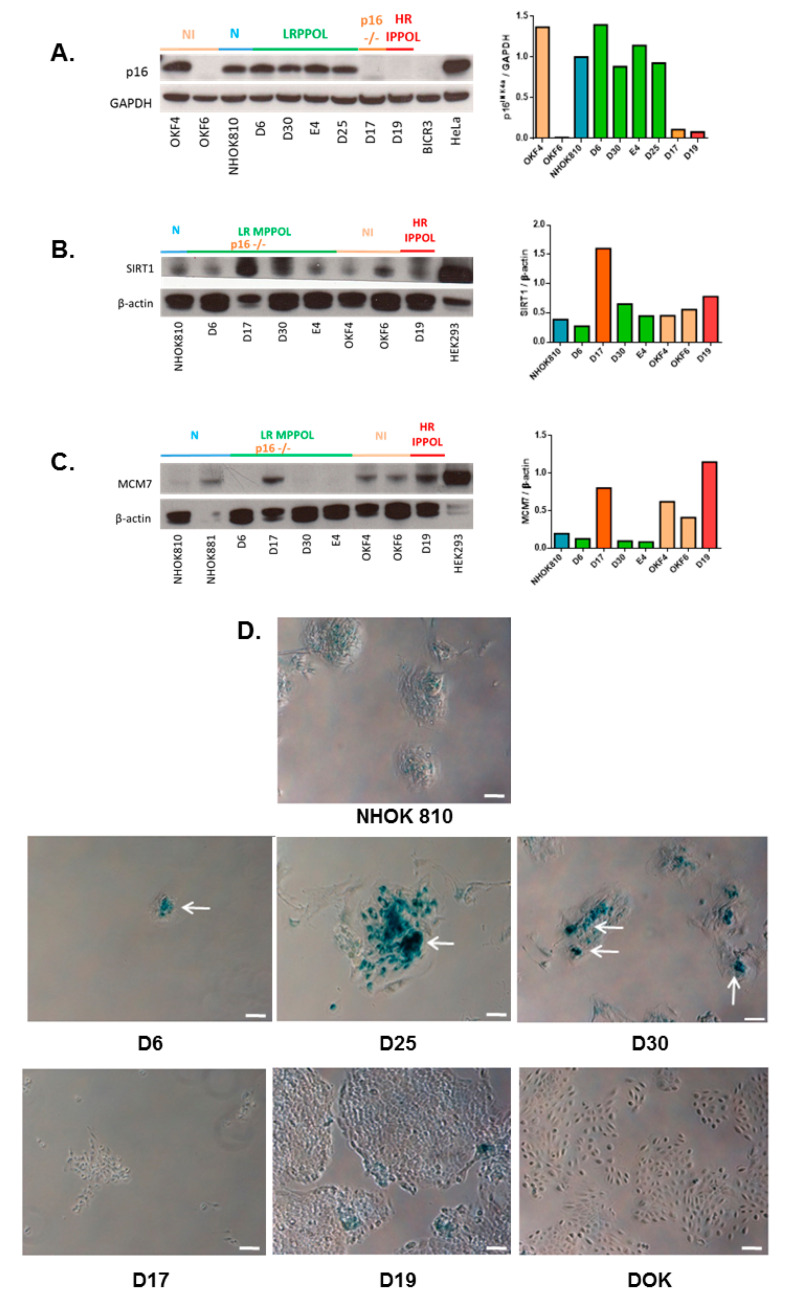
Characterisation of the senescence markers in the cell lines used in this study. Western blots of (**A**) p16^INK4A^ in normal keratinocyte NHOK810 (N—blue colour bar above Western blot images and in graph), normal oral keratinocytes immortalised by telomerase (NI—beige colour) retaining (OKF4) or lacking (OKF6) p16^INK4A^ expression, low-risk mortal potentially premalignant oral lesion D6, D25, D30, and E4 (LR MPPOL—green colour), LR MPPOL extended lifespan D17 (p16^INK4A^ −/− —orange colour), and a representative of high-risk immortal potentially premalignant oral lesion D19 (HR IPPOL—red colour). HeLa was used as a positive control and BICR3, which lacks p16^INK4A^ expression whilst retaining p14^ARF^ and p15^INK4B^ expression, was used as a negative control. GAPDH was used as a loading control. (**B**) SIRT1 in normal oral keratinocytes (NHOK810 and NHOK881), a subset of the LR MPPOL (D6, D30, and E4), HR IPPOL (D19), LR MPPOL p16^INK4A^ −/− line (D17) and normal oral keratinocytes immortalised by telomerase (OKF4 and OKF6). β-actin was used as a loading control. (**C**) MCM7 in subset of LR MPPOL (D6, D30, E4, and D25), HR IPPOL (D19), normal oral keratinocytes immortalised by telomerase (OKF4 and OKF6), normal oral keratinocyte (NHOK810) and LR MPPOL p16^INK4A^ −/− line (D17). HEK293 was used as a positive control for both MCM7 and SIRT1 and the symbols are the same as in (**A**). All lanes contain equal amounts of protein 20µg. Right panels represent quantification of p16^INK4A^, MCM7, and SIRT1 with histograms indicating mean ratios of the protein levels relative to GAPDH or β-actin loading control. All data is derived from one protein extract of each line. (**D**) SA-βGal staining of NHOK810, LR MPPOL (D6, D25, and D30), LR MPPOL p16^INK4A^ −/− line (D17), and HR IPPOL (D19 and DOK). Note the location of the blue cells indicated in the LR MPPOL cultures indicated by the white arrows in the centres of the colonies. White Bar = 100 µm see all panels). The whole Western blots are shown in the Appendix A.

**Figure 3 cancers-13-04212-f003:**
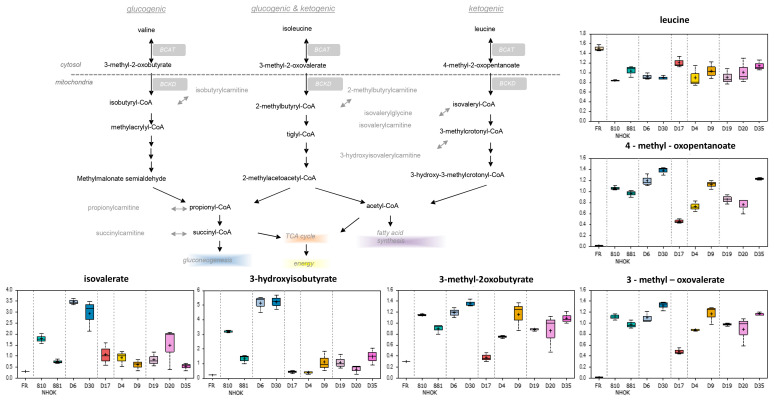
Alterations of the branch chain amino acid pathway in PPOL and normal keratinocytes. The normal keratinocytes are shown in green, the LR MPPOL keratinocytes in blue, the HR IPPOL keratinocytes in yellow (D4 and D9) and purple (D19, D20 and D35), and the LR PPOL p16^INK4A^ −/− line D17 in brown. Data represents the means of three independent experiments +/− standard deviation. For statistical analyses see Appendix A. BCAT = branched-chain amino acid transaminase; BCKD = Branched-chain alpha-keto acid dehydrogenase complex.

**Figure 4 cancers-13-04212-f004:**
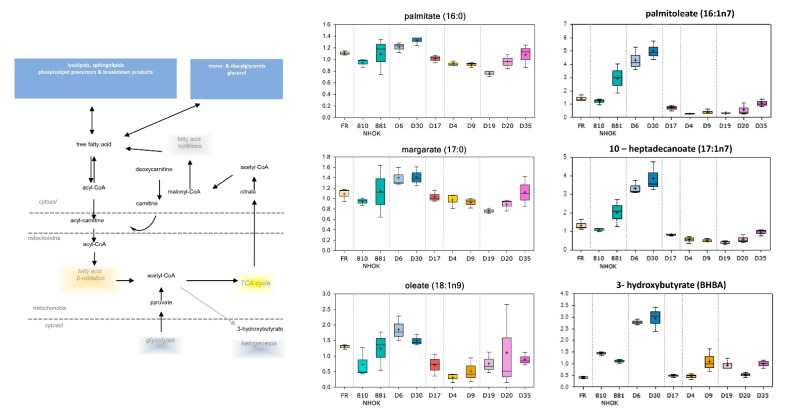
Alterations of lipid metabolism in PPOL and normal keratinocytes. The symbols are the same as for Figure 3. Data represents the means of three independent experiments +/− standard deviation. For statistical analyses see Appendix A. TCA = tricarboxylic acid cycle.

**Figure 5 cancers-13-04212-f005:**
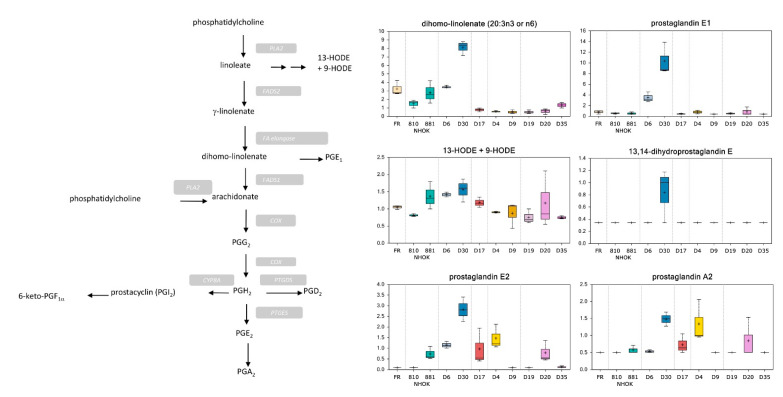
Alterations of prostaglandin metabolism in PPOL and normal keratinocytes. The symbols are the same as for Figure 3. Data represents the means of three independent experiments +/− standard deviation. For statistical analyses see Appendix A. COX = cyclooxygenase; FADS = fatty acid desaturase; PLA2 = phospholipase A2; FA = fatty acid; CYP8A = Cytochrome P450 8A; PTGDS = Prostaglandin D synthase; and PTGES = Prostaglandin E synthase.

**Figure 6 cancers-13-04212-f006:**
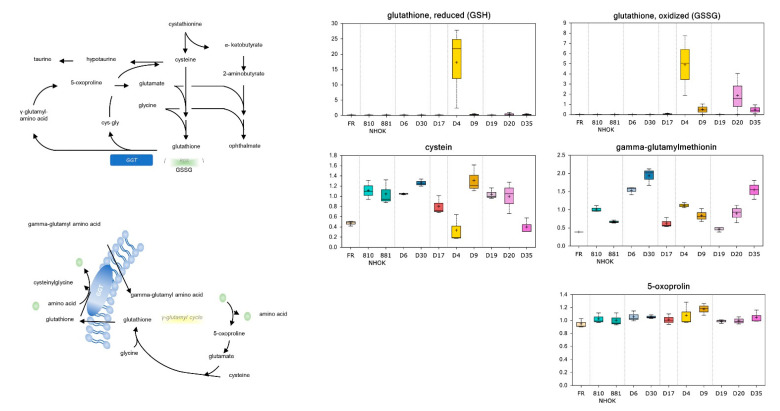
Alterations of the glutathione metabolism in PPOL and normal keratinocytes. The symbols are the same as for Figure 3. Data represents the means of three independent experiments +/- standard deviation. For statistical analyses see Appendix A. GGT = Gamma glutamyl transferase; GSH = reduced glutathione; and CSSG = oxidized glutathione.

**Figure 7 cancers-13-04212-f007:**
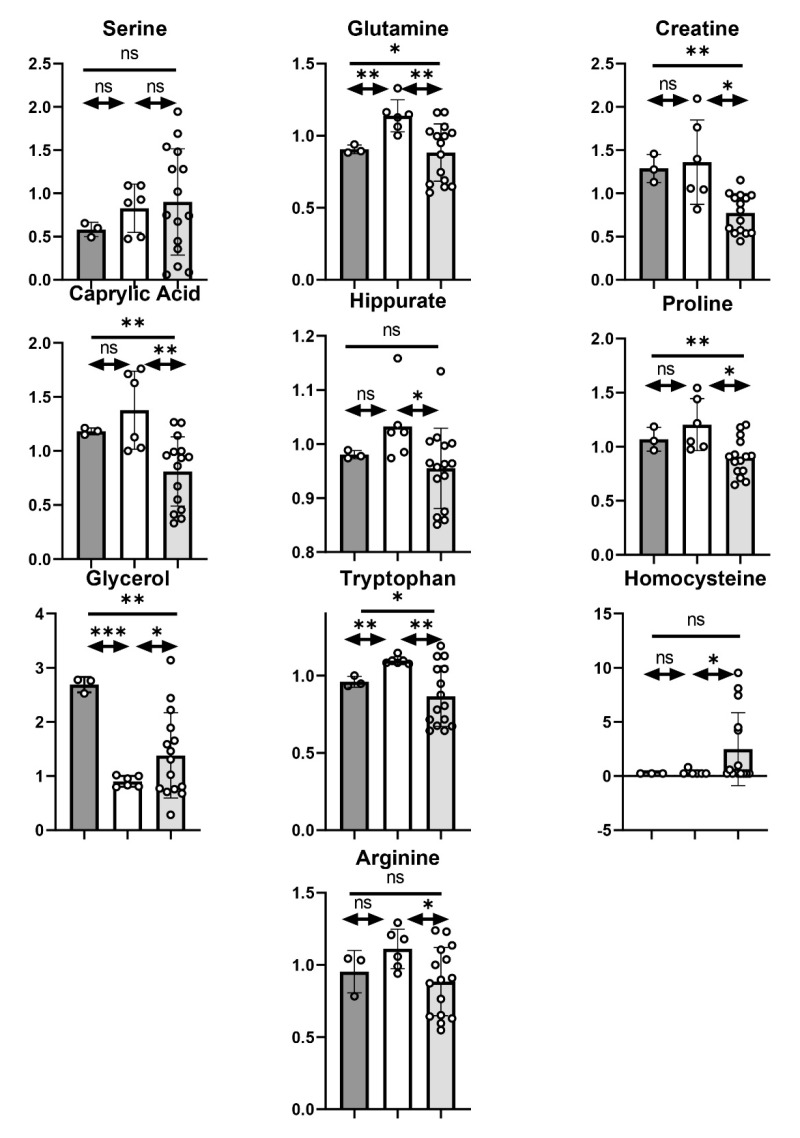
Extracellular LR PPOL and HR IPPOL keratinocyte metabolites distinguishing between PPOL and OSCC in saliva samples in vivo. The bar charts show selected metabolites from the unbiased metabolic screen that have been reported to be associated with PPOL progression in saliva samples in vivo, and which show similar trends distinguishing between LR MPPOL, HR IPPOL, and normal oral keratinocytes in vitro. The data are the results of three independent experiments +/− standard deviation derived from one line of healthy normal keratinocytes, two lines of LR MPPOL keratinocytes, and five lines of HR IPPOL keratinocytes. The data are not normalised for cell number or protein, but are from cultures of equal surface area. The data are presented as scaled intensity/mL of conditioned medium. Dark grey bars = normal oral keratinocytes; white bars = LR PPOL keratinocytes; and light grey bars = HR IPPOL keratinocytes. Significant by one-way ANOVA (bar) or Welch’s Test (arrowed lines) * *p* < 0.05; ** *p* < 0.01; and ***; *p* < 0.001.

**Figure 8 cancers-13-04212-f008:**
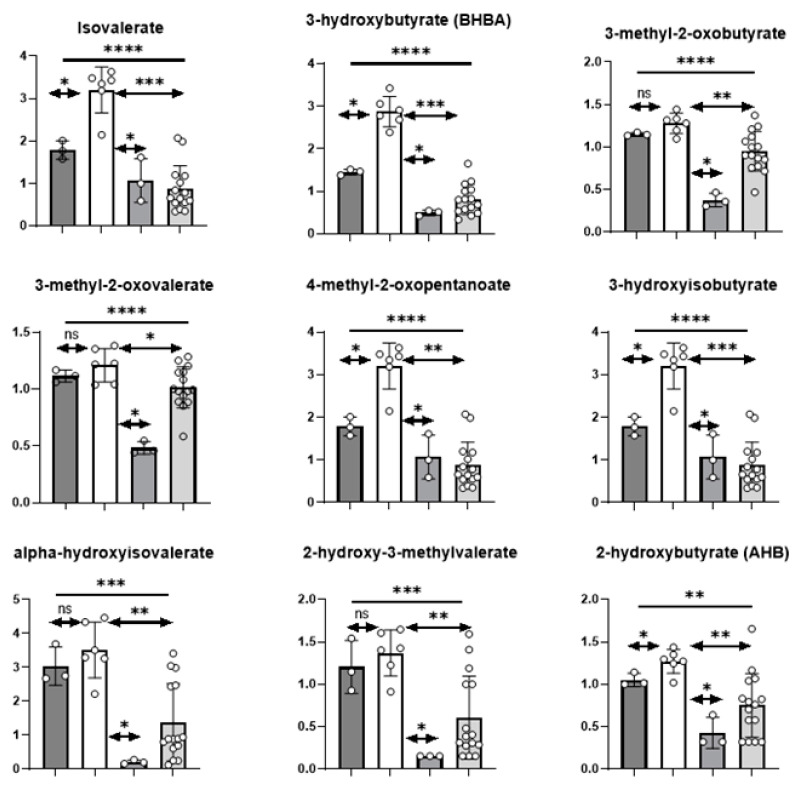
Volatile metabolites distinguishing LR MPPOL, HR IPPOL, and normal oral keratinocytes. The bar charts show selected volatile metabolites from the unbiased metabolic screen that distinguish LR MPPOL, HR IPPOL, and normal oral keratinocytes in vitro. The data are the results of three independent experiments +/− standard deviation derived from one line of healthy normal keratinocytes, two lines of LR MPPOL keratinocytes, D17, and five lines of HR IPPOL keratinocytes. The data are not normalised for cell number or protein, but are from cultures of equal surface area. Dark grey bars = normal oral keratinocytes; white bars = LR PPOL keratinocytes; lighter grey bars = D17, and very light grey bars = HR IPPOL keratinocytes. Significant by one-way ANOVA (bar) or Welch’s Test (arrowed Lines) * *p* < 0.05; ** *p* < 0.01; *** *p* < 0.001; and **** *p* < 0.0001.

**Figure 9 cancers-13-04212-f009:**
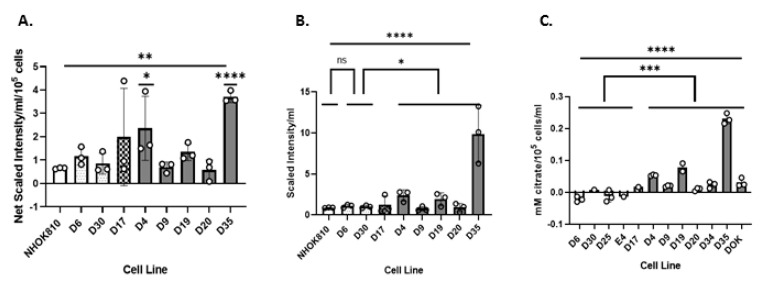
Levels of extracellular citrate are consistently elevated in HR IPPOL keratinocytes relative to LR MPPOL and normal keratinocytes. (**A**) The bar charts show citrate levels from the unbiased metabolic screen. The data are the results of three independent experiments +/− standard deviation derived from one line of healthy normal keratinocytes, two lines of LR MPPOL keratinocytes, D17 (p16^INK4A^ −/−), and five lines of HR IPPOL keratinocytes. The data are concentrations not normalised for cell number or protein, but are from cultures of equal surface area and presented as scaled intensity. White bar = normal oral keratinocytes; stippled bars = LR PPOL keratinocytes; hatched bars = D17 (p16^INK4A^ −/−) keratinocytes; and dark grey bars = HR IPPOL keratinocytes. Significant by one-way ANOVA (bar) or Welch’s Test relative to NHOK810 (bars over individual cell line bars) * *p* < 0.05; ** *p* < 0.01; and *** *p* < 0.001. (**B**) The data are the same as for (**A**), but with the background subtracted and normalised for cell number. The data are presented as net scaled intensity/10^5^ cells/mL. Significant by one way ANOVA (straight bar). Welch’s Test was used to compare LR MPPOL (D6 and D30) with HR IPPOL lines D4, D9, D19, D20, and D35 linked bars) * *p* < 0.05; ** *p* < 0.01; and *** *p* < 0.001. (**C**) The bar charts show citrate levels using targeted metabolomics analysis. The data are the results of between one (D30, E4, and D17) and three (rest) independent experiments +/− standard deviation derived from four lines of LR MPPOL keratinocytes, D17 (p16^INK4A^ −/−), and seven lines of HR IPPOL keratinocytes with the background subtracted and normalised for cell number. The data are presented as mM citrate/10^5^ cells/mL. Symbols are the same as in (**A**,**B**). Significant by one-way ANOVA (bar). Welch’s Test was used to compare LR MPPOL (D6, D25, E4, and D30) with HR IPPOL lines D4, D9, D19, D20, D34, DOK, and D35 linked bars) ** *p* < 0.01; *** *p* < 0.001; and **** *p* < 0.0001. D4 ***, D9 **, D20 **, D34 **, D35 ****, and DOK ** were all individually significant from D6 by the unpaired Student’s *t* test.

## Data Availability

(**A**) Materials: All cell lines will be made available subject to a reasonable request and the demonstration that the receiving laboratory has the means to maintain the cell lines successfully. Many lines are mortal but can be maintained using a 3T3 feeder layer and the Rho kinase inhibitor Y27632 as described in the methods section. The cell lines will be banked should funds become available. (**B**) Data sharing: All the raw data files used in the study will be made available on request. Unfortunately, Metabolon data is not acceptable for deposition in sites such as Metabolytes, as the company will not provide the necessary technical details.

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
