# Peer review of "The Extracellular Metabolome Stratifies Low and High Risk Potentially Premalignant Oral Keratinocytes and Identifies Citrate as a Potential Non-Invasive Marker of Tumour Progression"

_cancers, 2021, doi:10.3390/cancers13164212_

Round 1

Reviewer 1 Report

"The extracellular metabolome stratifies low and high risk potentially premalignant oral keratinocytes and identifies citrate as a potential non-invasive marker of tumour progression", by Karen-Ng et al is improved. Thank you for an interesting paper that should prove useful in understanding oral metabolite changes that are associated with high risk OSCC. The authors have addressed all my concerns.

Author Response

Thank you so much for your positive remarks on this manuscript.

Reviewer 2 Report

The author's amendments have significantly improved the manuscript.

However, several minor concerns remain.

  1. Figure 1 remains confusing. In a similar way to figure 2, please revise the legend in figure 1 to show which cell lines represent each group. For example, please change the legend to show that D6 and D30 are LR MPPOL etc.
  2. The text under the bar graphs and above the western blot images in Figure 2 are unclear and should be amended. Please include scale bars for all images shown in Figure 2C.
  3. A manuscript that is in preparation is unpublished. Please remove all references to unpublished work.
  4. Please revise the sentence on lines 680-683 from ‘However, even curable early stage cancer is expensive and as often associated with unacceptable morbidity’ to ‘However, even curable early stage oral cancer is expensive to treat and is often associated with unacceptable morbidity’.

Author Response

Kindly see attachment below. Thank you

Reviewer 3 Report

The authors have addressed all of my concerns and I believe this manuscript can be accepted.

Author Response

Thank you so much for your kind remarks on this manuscript.

This manuscript is a resubmission of an earlier submission. The following is a list of the peer review reports and author responses from that submission.

Round 1

Reviewer 1 Report

This manuscript presents interesting findings and results but needs extensive modification for grammar.

Examples:

Simple summary: : The early detection of mouth cancer is a high priority as improvements in this area could lead to greater cure rates and reduced disability due to extensive surgery. Mouth is very difficult to detect in over 70% of cases because it develops unseen until quite advanced and sometimes rapidly. Therefore, the development of markers in body fluids (liquid biopsies) indicative of sinister changes have a high priority. 

Do the authors mean "Mouth cancer" instead of "mouth"; Mouths are not difficult to detect. Perhaps oral cancer might be more appropriate, but the word cancer must not be left off. Also, changes are not generally "sinister", which typically describes a person or motive. Perhaps "oncogenic" or "cancerous" might be more appropriate. 

This manuscript has significant scientific value, which is not reflected by the quality of the writing that accompanies it. 

Example from Introduction: Senescence can be induced by a variety of cellular stresses but is bypassed in normal keratinocytes by the dual reduction of p53 and p16INK4A function and telomerase deregulation [2] and genetic alterations in all of these pathways are very common in OSCCs in vivo [3] and in immortal cell lines derived from potentially pre-malignant lesions (PPOL) [4] and OSCC [5].

It is not necessary to insert each reference inside the sentence, but rather should be left to the end of each sentence, such as "[2-5]"

Senescence can be induced by certain oncogenes [6] though not all [7] and this is sometimes referred to as oncogene-induced senescence. Senescent cells are present in human [8] and rodent [9, 10], pre-malignant lesions including oral and epidermal dysplasia [11] and in the oral cavity, senescence is almost ubiquitously bypassed upon progression to malignancy [3, 5, 11].

Although this information is technically correct, it does not flow well and does not add anything to our understanding of this particular manuscript. PPOL senescence genes are not the focus of this research study. Would recommend removal of entire paragraph.

The third paragraph in the introduction is far too long and disjointed. Each paragraph should have a topic sentence and 2-4 sentences that support the "topic". Please revise the introduction and discussion sections. 

Author Response

Kindly refer to the Word document.

Reviewer 2 Report

This manuscript describes the secreted metabolomes from a range of oral cell lines. It is claimed that several metabolites have been identified that can distinguish low risk MPPOL from high risk IPPOL and that many of these can also distinguish OSCC from PPOL in vivo. However, all work described in the manuscript was performed in an in vitro setting as far as this reviewer can tell.

MAJOR CONCERNS

1. Overall, the manuscript is poorly written and the key message is unclear. The apparent aim is to identify metabolites as potential liquid biopsy biomarkers but the discussion focuses more on the hypothetical function of the metabolites that were identified. The problem is that (almost) no functional studies were performed and this makes the paper confusing. 

2 .Unpublished work has been cited repeatedly throughout the manuscript and is unhelpful.  

3. Please clarify why informed consent was deemed not applicable despite stating that “all the patients were consented prior to biopsies being placed in cell culture”. It is appreciated that “all of the keratinocyte cultures used in this study were derived prior to 2002 and have been passaged and so are deemed cell lines and exempt from The Human Tissue Act of 2004” but if informed consent was obtained, then please provide institutional approval for this consent.

4. Considering that cancers in general are heterogeneous and are comprised of mixed cell types, profiling metabolites from single, simplified cells lines that fail to differentiate is unlikely to identify clinically relevant and translational results. 

5. The data showing senescence should be included in the manuscript.

6. In the discussion, the statement “very early stage cancer of the tongue needs extensive and expensive surgery to achieve long-term cure” is inaccurate and inconsistent with recent literature. In fact, early stage tongue cancers do very well based on improved surgical techniques.

7. The clinical details for each cell line were not provided. Clinical details such as grade, stage, alcohol and smoking status, age and gender are important and relevant. These parameters may confound the results but appear to have not been considered.

MINOR CONCERNS

1. Figure 1 is confusing in its current state and too many acronyms have been used. Please group cells from similar anatomical locations together using similar colours. Using D6 etc as names for cells lines makes the data difficult to understand. Also, the authors refer to cell type but this is not helpful. Please refer to anatomical location. The use of ‘mouth cancer’ to describe cancers the occur in the oral cavity, base of tongue is inappropriate. The head and neck cancer community will not appreciate this loose terminology.

2. The simple summary needs significant revising. The language and tone are strange. The conclusions are overstated.

3. The discussion is incomplete and lacks context in most paragraphs.

Reviewer 3 Report

"The extracellular metabolome stratifies low and high risk potentially premalignant oral keratinocytes and identifies citrate as a potential non-invasive marker of tumour progression," by Karen Ng et al. is an interesting paper that takes advantage of cell strains and lines that were originally isolated from premalignant oral lesion some of which are High risk and the other low risk, along with two normal oral kertioncyte cell strains and some tumor lines. This is a fascinating study that could be done by few other groups. It should be noted due to the limited number of samples and the lack of correction from multiple testing with the welches T test in most comparisons,  the work is preliminary in nature, unless the authors had a limited number of metabolites they chose at the start to focus on. The authors are correct to mention difficulties in separating out characteristics due to inflammation from premalignancy.

Line 99

Signatures of PPOL and OSCC have been reported 95 recently [31] but it is not yet clear how many of these metabolites are related to the PPOL 96 keratinocytes, inflammatory disease [32] or bacterial breakdown products of other metabolites and changes in the oral microbiome are known to occur in PPOL and OSCC [34].

  • Writing unclear in this sentence

Line 168

The scaled intensities were either not normalized or normalized to account for the variation in biomass or cell number between 170 different experimental groups or were normalized for cell number following subtraction 171 of the medium blank values and expressed as net scaled intensity per 105 cells per ml of  conditioned medium.

It is unclear why a single method of noramlisation was not used.

Line 386

Interestingly, some metabolites recently reported to distinguish PPOL and OSCC in 387 saliva in vivo [31] also showed similar trends between HR IPPOL and LR MPPOL keratino-388 cytes. These included elevated 4-hydroxybutyrate, serine, glutamate, glycerol, leucine and 389 depleted hippurate, proline, glycerol-3-phosphate, caprylate, histidine, glycerophos-390 phoryl choline, arginine tryptophan, creatine and phenylalanine distinguished HR IPPOL 391 from LR MPPOL in vitro with some of these were significantly different (Figure 6).

Are these similarities statistically significant? 12 chemicals differ the same way in both systems these cells and in saliva. . But how many metabolites are different between PPOLO and OSCC in saliva and HR IPPOL and LR MPPOL?

Given the need to remove the 3T3 feeder layer prior to the assays, there needs to be an explanation of why alternative growth condition without serum and without the feeder layer were not used sucha s those described in the paper below.

 Cancer Discov. 2019 Jul;9(7):852-871. doi: 10.1158/2159-8290.CD-18-1522. Epub 2019 May 3.Oral Mucosal Organoids as a Potential Platform for Personalized Cancer Therapy

I am confused. I thought NHOK810 and NHOK881 were primary oral keratinocytes. In supp table S1A are all cells true cells lines that are immortal? For example it seems that D6 and D30 are not immortal.This should be clearly makred in supplemental table 1a

Supp table 3a What does “P value  (Welch’s T test) Corrected for Cell Number”, mean?. Do you mean corrected for number of cells in the plate? If so is that the right place for that information?

Supplementary Table  5a- There are only two normal cell strains as far as I can cell. How many technical replicates were there for the comparisons done? I could not easily find that information.
